# Cohort study of neonatal resuscitation skill retention in frontline healthcare facilities in Bihar, India, after PRONTO simulation training

Brennan V Higgins ,[1] Melissa M Medvedev,[1,2] Hilary Spindler,[3] Rakesh Ghosh,[3] Ojungsangla Longkumer,[4] Susanna R Cohen,[5] Aritra Das,[6] Aboli Gore,[6] Tanmay Mahapatra,[6] Dilys M Walker[7]

For numbered affiliations see end of article.

**Correspondence to**
Dr Brennan V Higgins; brennan.vail@ucsf.edu

## ABSTRACT

**Background** Use of simulation in neonatal resuscitation (NR) training programmes has increased throughout low-income and middle-income countries. Many of such programmes have demonstrated a positive impact on NR knowledge and skill acquisition along with reduction of early neonatal mortality and fresh stillbirth rates. However, NR skill retention after simulation programmes remains a challenge.

**Methods** This study assessed facility level NR skill retention after PRONTO International's simulation training in Bihar, India. Training was conducted within CARE India's statewide in-job, on-site Apatkaleen Matritva evam Navjat Tatparta mentoring programme as part of a larger quality improvement and health systems strengthening initiative. Public sector facilities were initially offered training, facilitated by trained nursing graduates, during 8-month phases between September 2015 and January 2017. Repeat training began in February 2018 and was facilitated by peers. NR skills in simulated resuscitations were assessed at the facility level at the midpoint and endpoint of initial training and prior to and at the midpoint of repeat training.

**Results** Facilities administering effective positive pressure ventilation and assessing infant heart rate increased (31.1% and 13.1%, respectively, both p=0.03) from midinitial to postinitial training (n=64 primary health centres (PHCs) and 192 simulations). This was followed by a 26.2% and 20.9% decline in these skills respectively over the training gap (p≤0.01). A significant increase (16.1%, p=0.04) in heart rate assessment was observed by the midpoint of repeat training with peer facilitators (n=45 PHCs and 90 simulations). No significant change was observed in other skills assessed.

**Conclusions** Despite initial improvement in select NR skills, deterioration was observed at a facility-level post-training. Given the technical nature of NR skills and the departure these skills represent from traditional practices in Bihar, refresher trainings at shorter intervals are likely necessary. Very limited evidence suggests peer simulation facilitators may enable such increased training frequency, but further study is required.

## INTRODUCTION

Neonatal mortality represents a disproportionate and unsolved public health

### What is known about the subject?

► Neonatal resuscitation simulation training programmes have spread throughout low-income and middle-income settings, most with a model of brief periods (1–3 days) of instruction.
► These training programmes can improve provider knowledge and skill and thereby reduce early neonatal mortality and fresh stillbirths.
► Skill retention after such training programmes has been achieved only in programmes with ongoing opportunities for training and practice.

### What this study adds?

► This study evaluates skill retention after in-job, on-site nurse mentoring with simulation training carried out over an 8-month period with the hypothesis that a longer duration of training may improve retention.
► Despite a longer duration of initial training, skill deterioration was still observed post-training.
► Refresher trainings were possible with peer simulation facilitators but demonstrated limited skill improvement among nurse mentees.

problem globally,[1] and intrapartum-related events remain a leading cause.[2] In response, neonatal resuscitation (NR) simulation training programmes have proliferated in low-income and middle-income countries (LMICs).[3–5] Nevertheless, lack of skills among frontline providers remain a well-recognised barrier to immediate neonatal care[6] and a focus of the Every Newborn Action Plan, which aims to reduce neonatal mortality to 10 per 1000 live births in every country globally by 2035.[7] Although NR training programmes have already had a documented positive impact on knowledge and skill acquisition with subsequent reduction in early neonatal mortality and fresh stillbirths, skill retention

after these programmes remains a unique challenge and a key obstacle to the 2035 goal.[3–5]

In LMICs, studies evaluating skill retention after NR simulation training programmes have assessed retention from 1 month to 12 months post-training. The majority of these studies assessed Helping Babies Breathe (HBB), a 1–3 day simulation-based training programme developed specifically for LMIC settings.[8] Other simulation training programmes assessed were of comparable duration and content to HBB. Most of these studies reported some decline in NR skills after training completion, including at 1 month,[9] 3 months,[10] 4 months[11] and 6 months[12–14] post-training.

Notably, however, three studies did not observe uniform skill decay. One study observed maintenance of skills at 9–12 months post-training, which the authors attributed to the unique study setting—a small group of providers (n=14) in a high-risk referral hospital in Accra, Ghana, where providers perform NR daily.[15] A second study found a falloff in skills in two of three training groups at 3 to 6 months but maintenance of skills in the third group at this same timepoint.[14] In this latter group, simulators were left at the study facilities for ongoing practice, which was facilitated by clinical mentors who received phone support from district trainers. A final study saw improvements at 3 month and 12 months post-training in all skills except positive pressure ventilation (PPV).[16] In this study, facilities received the initial HBB training (2 days) in addition to biweekly visits from trainers to encourage practice over the post-training period (12 months). Common to all three studies was the opportunity for frequent and ongoing practice. However, PPV was highlighted as an exception, with an observed decline in competence despite ongoing practice.[16]

PRONTO International[17] developed a unique in situ, intrapartum simulation training curriculum for LMIC settings, which was implemented phase-wise in Bihar, India, over a long initial training period of approximately 8 months per phase. The training was embedded in a state-wide mentoring programme called 'Apatkaleen Matritva evam Navjat Tatparta' (AMANAT) and implemented by CARE India in partnership with Government of Bihar. Assessment of PRONTO training implemented with AMANAT mentoring has demonstrated improved NR skills post-training in both simulated and live deliveries,[18] despite the presence of many structural, logistical and cultural barriers to neonatal care.[19 20] However, NR skill retention following PRONTO training, or any simulation training of comparable duration in an LMIC setting, has not been assessed. The aim of this study was to assess retention of NR skills after PRONTO simulation training within the AMANAT mentoring programme in Bihar for programmatic evaluation and possible application in similar LMIC settings. A secondary objective of this study was to assess skill reacquisition with peer-facilitated simulations to better understand the feasibility of this approach to simulation training.

## METHODS
### Study setting
Training was conducted in Bihar, the third most populous state in the country of India.[21] According to the World Bank, more than one-third of Bihar's population lives below the international poverty line (<$1.90 per day).[22] The documented neonatal mortality rate is 37 per 1000 live births,[23] with significant under-reporting likely. This study assessed PRONTO training conducted alongside AMANAT mentoring in government basic emergency obstetric and neonatal care facilities that were largely primary health centres (PHCs).

PHCs serve as entry-level facilities for preventative and basic medical care. Approximately 1.3 million deliveries are conducted annually in PHCs.[24] The delivery load per facility varies by location. Approximately 10% of mothers are referred from PHCs to higher level facilities for delivery.[25] Deliveries in PHCs are attended by auxiliary nurse midwives (ANMs) or general nurse midwives (GNMs), frontline providers with 2 or 3.5 years of training after general education, respectively.[26] ANMs/GNMs perform a variety of tasks at PHCs in addition to labour and delivery care. These often include other clinical obligations, including emergency care and outpatient care, as well as other tasks such as taking maintaining inventory supply and completing maternity registers. The average out-of-pocket expenditure of mothers in 2019 for a PHC delivery was 1788 Indian rupees, equivalent to US$25. The majority of expenses resulted from diagnostic tests.[25]

### AMANAT and PRONTO interventions
AMANAT was a state-wide in-job, on-site mentoring programme for obstetric and newborn care providers in public sector facilities. The programme was implemented by CARE India[27] in collaboration with the government of Bihar as part of a health system strengthening and quality of care improvement effort. PRONTO International[17] provided simulation training on obstetric and neonatal emergencies within the broader AMANAT initiative using the AMANAT mentor/mentee structure for clinical instruction. PRONTO training consists of in situ, intrapartum simulations in addition to teamwork and communication exercises, skill stations and case-based learning. The intrapartum nature of PRONTO training emphasises care of the mother–infant dyad, including complications that may coexist such as NR and postpartum haemorrhage.[28]

This study assessed PRONTO training conducted within the AMANAT initiative at PHCs between September 2015 and February 2019. The timeline of the intervention is depicted in figure 1. From September 2015 to January 2017, mentoring and training were conducted in three equivalent 8-month phases (AMANAT phases 2–4). AMANAT phase 1 was excluded from this analysis due to curricular and simulation assessment differences during that initial phase. Each phase reached 80 non-overlapping PHCs for a total of 240 PHCs. This was followed by a training gap prior to initiation of repeat

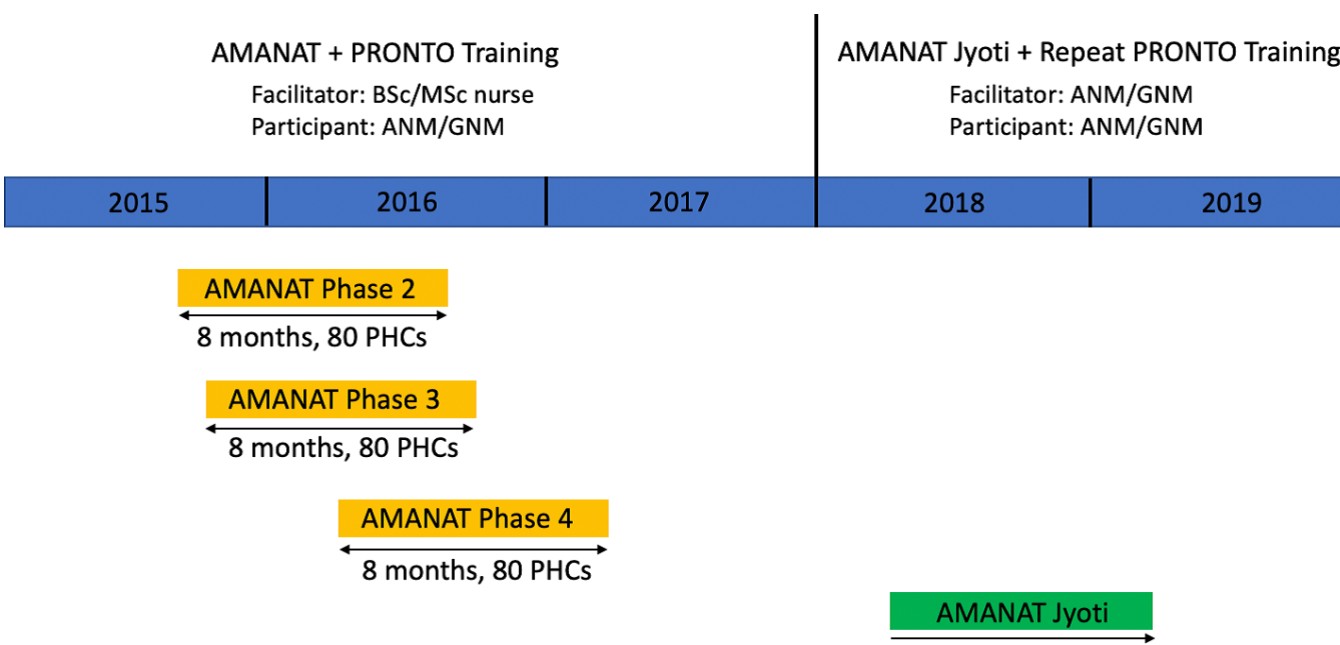

**Figure 1** Timing of AMANAT and AMANAT Jyoti training phases. AMANAT, Apatkaleen Matritva evam Navjat Tatparta; ANM, auxiliary nurse midwives; GNM, general nurse midwives; PHC, primary health centre.

PRONTO training as part of a second chapter of the AMANAT initiative called AMANAT Jyoti, which is a peer-guided mentoring programme aimed at sustainability. AMANAT Jyoti is currently ongoing. This second round of PRONTO training within AMANAT Jyoti reached 353 PHCs state-wide. The majority of these PHCs had received prior training/mentoring during AMANAT and thus AMANAT Jyoti marked the initiation of repeat PRONTO training. However, some of the PHCs where AMANAT Jyoti with refresher PRONTO training was rolled out were not included in the earlier PRONTO training and AMANAT mentoring programme. Furthermore, even if the PHC had received prior simulation training and mentoring, due to staff turnover, not all ANMs/GNMs in a given PHC at the time of AMANAT Jyoti had participated in the prior training and mentoring. The exact percentage of ANMs/GNMs who had received prior training is unknown.

During the AMANAT phase of the programme, PRONTO NR simulations along with AMANAT mentoring were facilitated by nurses who were college educated (BSc and MSc) and recruited from across India. Mentees who received the simulation curriculum and mentoring were ANMs and GNMs posted in labour rooms in PHCs. ANM/GNMs were not paid to participate in PRONTO training or AMANAT mentoring, and this was their first experience with simulation-based NR training. Some ANM/GNM mentees had participated in other governmental classroom-based trainings on topics including basic newborn care and NR. To ensure that AMANAT Jyoti had a sustainability model built into it, mentors, and thus simulation facilitators, transitioned to a cadre of trained ANMs and GNMs working in the Bihar PHC system. Mentees were peer ANM/GNMs.

## Simulations and assessments

In AMANAT, nurse mentors worked in pairs under the supervision of a master nurse mentor to facilitate PRONTO's NR simulations at four PHCs, spending 1 week per month mentoring at each PHC. As depicted in figure 1, NR simulations were conducted beginning in month 3 of mentoring and continued monthly through the end of the mentoring phase (month 8). NR clinical instruction during live deliveries began at the outset of the phase.

A single NR assessment simulation was conducted to assess skills at a facility level at month 4 ('AMANAT midassessment') and month 8 ('AMANAT postassessment') of training. Individual participants in the assessment simulation were chosen at random. Pretraining assessments were not conducted to give mentees time to adjust to simulation-based learning and thereby minimise the potential impact of simulation artefact on data. In AMANAT phases 2 and 3, the NR assessment scenario began immediately after birth, whereas in phase 4, the scenario also included a normal spontaneous vaginal delivery.

Prior to AMANAT Jyoti, a single NR simulation ('AMANAT Jyoti baseline reassessment') was conducted in a random sample of PHCs (n=105) to assess skill retention at the facility level from AMANAT postassessments. Following baseline reassessments, peer NR simulation training along with mentoring was then introduced between July and October 2018. One simulation was conducted per PHC during this interval. Subsequently, AMANAT Jyoti mid reassessments were conducted

between October 2018 and February 2019 to assess NR skill acquisition, again at a facility level, with the new peer mentorship model. Both AMANAT Jyoti baseline reassessment and mid reassessment simulations began with a normal spontaneous vaginal delivery and subsequently required NR.

All assessment simulations were video-recorded for immediate debriefing and programme assessment purposes.

### Ethics approval

All mentees provided consent for the use of video data in aggregated analyses.

### Patient and public involvement

The data for this manuscript were simulation data, and participants were frontline healthcare providers. Patients did not participate in this research. Results will be disseminated to local facilities through data dissemination efforts conducted collaboratively by PRONTO International, CARE India and the Government of Bihar officials to assist with quality of care improvement efforts.

### Analysis

NR assessment simulation videos from all phases were analysed by a Neonatal Resuscitation Program[29] certified physician at UCSF. Due to the minor differences in the simulated clinical scenario (with or without a preceding delivery), data could not be coded in a single-blinded group. Assessment data were matched by facility across time to assess NR skill acquisition and retention at the facility level. The four assessment timepoints per PHC were AMANAT midassessments, AMANAT postassessments, AMANAT Jyoti baseline reassessments and AMANAT Jyoti mid reassessments. The training gap between AMANAT postassessments and AMANAT Jyoti baseline reassessments allowed for assessment of NR skill retention at a facility level. The duration of this gap ranged from 15 months to 29 months depending on the AMANAT phase in which the PHC originally participated (figure 1).

A Cochran's Q test was conducted to assess if there was any difference across matched facilities in the percentage of facilities in which key NR skills were correctly performed at any time from AMANAT midassessments to AMANAT Jyoti baseline reassessments. Due to the matched nature of this test, only facilities where skills could be evaluated at all timepoints (midassessment, postassessment and baseline reassessment) were included in the statistical analysis. The number of facilities included varied by skill as not all skills could be evaluated in every simulation video due to video quality issues. For skills that demonstrated a statistically significant change over time, McNemar's test was used to individually assess change from AMANAT midassessments to postassessments, AMANAT postassessments to AMANAT Jyoti baseline reassessments and AMANAT midassessments to AMANAT Jyoti baseline reassessments. Additionally, a McNemar's test was conducted to assess

skill acquisition at the facility level during AMANAT Jyoti from baseline reassessments to mid reassessments.

We also fit mixed effect models with a random intercept for each facility to minimise loss of data due to missing timepoints. Data were available on each indicator at several, but not all, timepoints. Individual observations were clustered within facilities, giving it a two-level data structure. As the indicators were dichotomous, we used a logistic regression model to estimate the ORs. Simulations were performed once monthly during the initial AMANAT training, hence the associations are reported as an increase in the odds of the individual skills for every month of simulation training. We used the '*melogit*' programme in Stata to fit the models with a random intercept term for facility. Statistical analyses were conducted using SPSS V.23[30] and Stata V.16.[31]

## RESULTS
### NR skill retention

PRONTO NR simulations were conducted at all three initial timepoints including AMANAT midassessments, AMANAT postassessments and AMANAT Jyoti baseline reassessments at 64 PHCs across Bihar. This included 24 PHCs initially trained in AMANAT phase 2, 14 in AMANAT phase 3 and 26 in AMANAT phase 4.

A significant increase in the percentage of facilities in which two key NR skills—PPV with chest rise and heart rate assessment—were performed correctly was observed over the initial training period (AMANAT mid to post) followed by a decline over the training gap (AMANAT post to AMANAT Jyoti baseline; table 1, figure 2, p<0.01). For PPV with chest rise, a 31.1 percentage point increase from midinitial to postinitial training was observed (p=0.03), followed by a 26.2 percentage point decrease over the training gap (p=0.01). For assessment of heart rate, an initial 13.1 percentage point increase from midtraining to post-training (p=0.03) was followed by a 20.9 percentage point decrease over the training gap (p<0.01).

In all other skills assessed including stimulation, suction, neck extension and rate of PPV delivery, there was no statistically significant changes over time. Finally, there was no statistically significant change from midassessments to baseline reassessments for all skills including PPV with chest rise and heart rate assessment.

The findings above were confirmed by multilevel models, where convergence was achieved (table 2). From AMANAT mid to post, there was an increased in odds of correctly performing PPV with chest rise per month of simulation training (OR 1.50, 95% CI 1.15 to 1.96, p<0.01). However, from AMANAT post to AMANAT Jyoti baseline when there was a gap in training, a lower odds per month passed was observed (OR 0.91, 95% CI 0.86 to 0.97, p<0.01). Results for heart rate assessment were similar for the time points AMANAT mid to AMANAT Jyoti baseline, where a small decrease in odds of performing this skill was observed (OR 0.95, 95% CI 0.91 to 0.99, p=0.04).

**Table 1** Percentage of facilities in which key skills were correctly performed in simulated NRs at mid-PRONTO training, post-training and following a 15–29 month training gap (n=192 assessment simulations at 64 facilities)

| NR skill | N* | AMANAT mid assessment† % (n)¶ | AMANAT post assessment‡ | AMANAT Jyoti baseline re-assessment | P value** | mid to post§ Δ % | post to baseline§ | mid to baseline§ |
|---|---|---|---|---|---|---|---|---|
| Stimulation | 60 | 55.6 (35) | 50.8 (31) | 55.6 (35) | 0.92 | −4.8 | 4.8 | 0 |
| Suction | 58 | 63.9 (39) | 67.8 (40) | 67.2 (43) | 0.82 | 3.9 | −0.6 | 3.3 |
| Neck extension | 49 | 78.2 (43) | 80.7 (46) | 80.6 (50) | 0.68 | 2.5 | −0.1 | 2.4 |
| PPV with chest rise | 41 | 58.9 (33) | 90.0 (45) | 63.8 (37) | 0.01 | 31.1* | −26.2* | 4.9 |
| Chest rise in ≤30 s | 14 | 53.1 (17) | 51.2 (22) | 68.6 (24) | 0.46 | −1.9 | 17.4 | 15.5 |
| Rate of PPV 40–60 bpm | 53 | 36.7 (22) | 41.4 (24) | 38.7 (24) | 0.88 | 4.7 | −2.7 | 2.0 |
| Heart rate assessed | 54 | 85.2 (52) | 98.3 (57) | 77.4 (48) | <0.01 | 13.1* | −20.9* | −7.8 |

*Total number of facilities in which NR skill could be assessed at all three timepoints mid through baseline; n varies due to video quality.
†Includes AMANAT phases 2, 3 and 4 midtraining assessments.
‡Includes AMANAT phases 2, 3 and 4 post-training assessments.
§Significant change based on McNemar's test; p values reported in text.
¶Per cent (number) of facilities in which NR skill was performed correctly at a given timepoint.
**Cochran's Q test.
AMANAT, Apatkaleen Matritva evam Navjat Tatparta; BPM, beats per minute; NR, neonatal resuscitation; PPV, positive pressure ventilation.

## PRONTO Training Structure during AMANAT

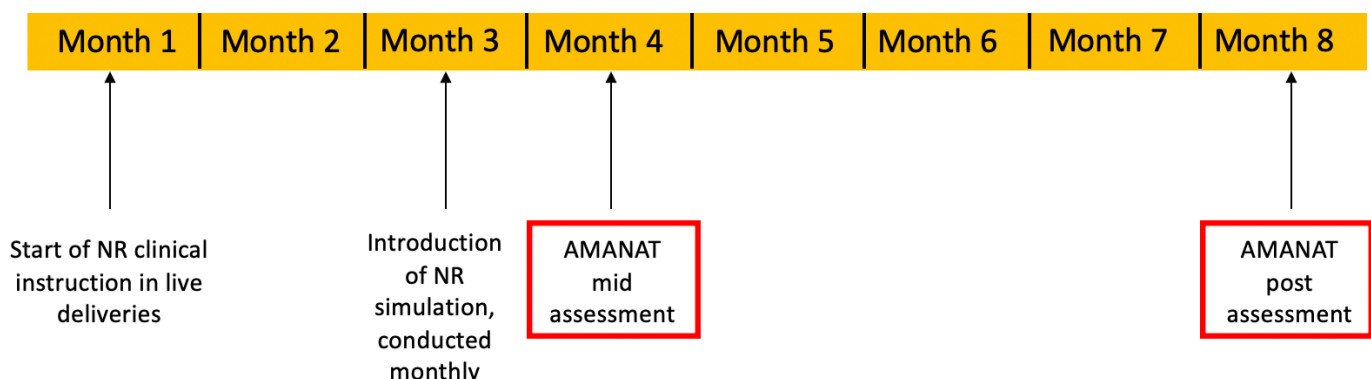

## PRONTO Training Structure during AMANAT Jyoti

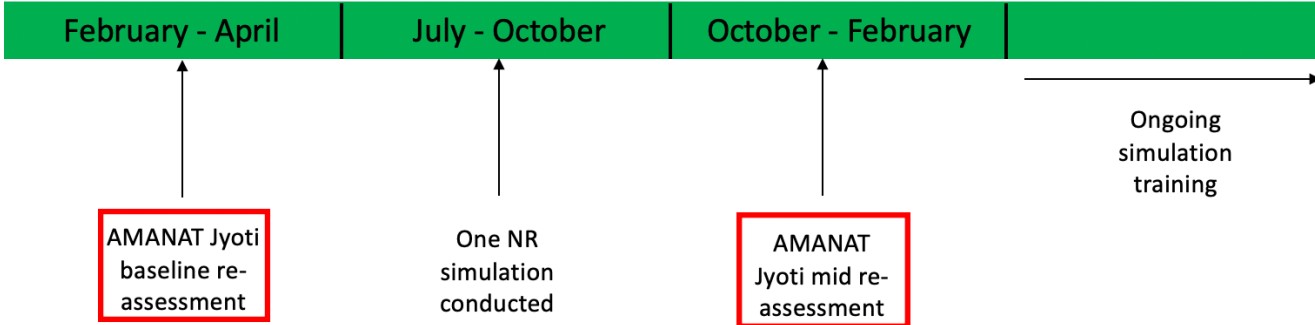

**Figure 2** PRONTO training timeline and structure within AMANAT and AMANAT Jyoti. AMANAT, Apatkaleen Matritva evam Navjat Tatparta; NR, neonatal resuscitation.

**Table 2** Mixed effects model of facility-based NR skill change with PRONTO training during AMANAT and AMANAT Jyoti

| NR skill | AMANAT mid* to AMANAT post† OR‡ (95% CI) | P value | AMANAT post† to AMANAT Jyoti baseline OR‡ (95% CI) | P value | AMANAT mid* to AMANAT Jyoti baseline OR‡ (95% CI) | P value | AMANAT Jyoti baseline to AMANAT Jyoti mid OR‡ (95% CI) | P value |
|---|---|---|---|---|---|---|---|---|
| Stimulation | 0.84 (0.67 to 1.06) | 0.15 | 1.00 (0.97 to 1.04) | 0.91 | 1.00 (0.97 to 1.04) | 0.83 | 1.00 (0.91 to 1.09) | 0.96 |
| Suction | 1.02 (0.83 to 1.25) | 0.87 | 0.98 (0.94 to 1.02) | 0.38 | 0.99 (0.96 to 1.03) | 0.75 | 0.96 (0.90 to 1.03) | 0.22 |
| Neck extension | 1.04 (0.81 to 1.33) | 0.76 | 1.00 (0.96 to 1.04) | 0.98 | 1.00 (0.97 to 1.04) | 0.98 | 1.03 (0.92 to 1.15) | 0.64 |
| PPV with chest rise | 1.50 (1.15 to 1.96) | <0.01 | 0.91 (0.86 to 0.97) | <0.01 | 0.98 (0.94 to 1.01) | 0.19 | 1.01 (0.90 to 1.13) | 0.83 |
| Chest rise in ≤30 s | 1.01 (0.79 to 1.28) | 0.96 | 1.05 (0.98 to 1.12) | 0.15 | 1.04 (0.99 to 1.09) | 0.11 | 0.90 (0.81 to 1.00) | 0.05 |
| Rate of PPV 40–60 bpm | 1.13 (0.91 to 1.42) | 0.27 | 0.98 (0.95 to 1.02) | 0.30 | 0.99 (0.96 to 1.02) | 0.61 | 1.00 (0.93 to 1.08) | 0.90 |
| Heart rate assessed | § | | § | | 0.95 (0.91 to 1.00) | 0.04 | 1.22 (1.00 to 1.48) | 0.05 |

*Includes AMANAT phases 2, 3 and 4 midtraining assessments.
†Includes AMANAT phases 2, 3 and 4 post-training assessments.
‡Change in odds of the individual skill per month.
§Models did not converge.
AMANAT, Apatkaleen Matritva evam Navjat Tatparta; BPM, beats per minute; NR, neonatal resuscitation; PPV, positive pressure ventilation.

**Table 3** Percentage of facilities in which key skills were correctly performed in simulated NR at baseline and mid-repeat PRONTO training with peer simulation facilitators (n=90 assessment simulations at 45 facilities)

| NR skill | N* | AMANAT Jyoti baseline re-assessment % (n)† | AMANAT Jyoti mid re-assessment | baseline to mid Δ % | P value‡ |
|---|---|---|---|---|---|
| Stimulation | 44 | 52.3 (23) | 62.2 (28) | 9.9 | 0.45 |
| Suction | 45 | 71.1 (32) | 66.7 (30) | −4.4 | 0.84 |
| Neck extension | 43 | 77.3 (34) | 90.9 (40) | 13.6 | 0.11 |
| PPV with chest rise | 43 | 65.1 (28) | 77.8 (35) | 12.7 | 0.15 |
| Chest rise in ≤30 s | 21 | 65.4 (17) | 46.9 (15) | −18.5 | 0.11 |
| Rate of PPV 40–60 bpm | 43 | 41.9 (18) | 51.1 (23) | 9.2 | 0.38 |
| Heart rate assessed | 44 | 79.5 (35) | 95.6 (43) | 16.1 | 0.04 |

*Total number of facilities in which NR skill could be assessed at both timepoints; n varies due to video quality.
†Per cent (number) of facilities in which NR skill was performed correctly at a given timepoint.
‡McNemar's test.
AMANAT, Apatkaleen Matritva evam Navjat Tatparta; BPM, beats per minute; NR, neonatal resuscitation; PPV, positive pressure ventilation.

## NR skill reacquisition with peer simulation facilitators

Of the 64 facilities that completed NR assessments at all three initial timepoints, 40 had a fourth NR assessment simulation at the midpoint of AMANAT Jyoti. Additionally, five facilities that received simulation training and mentoring for the first time during AMANAT Jyoti participated in both baseline reassessment and mid reassessment. Among these 45 facilities, a significant increase was observed in the percentage of facilities in which heart rate was assessed from baseline (79.5%) to mid (95.6%) reassessment (table 3, p=0.04). There were no statistically significant changes in the remaining skills assessed including stimulation, suction, neck extension, PPV with chest rise and rate of PPV delivery (table 3).

Multilevel modelling also demonstrated a higher odds of heart rate assessment from AMANAT Jyoti baseline to AMANAT Jyoti mid reassessment (table 2, OR 1.22, 95% CI 1.00 to 1.48, p=0.05). There was no statistically significant change in the odds of performing PPV with chest rise for these two timepoints. However, of those facilities where chest rise was achieved, there was a small decrease in odds of it being achieved within ≤30 s (table 2, OR 0.90, 95% CI 0.81 to 0.99, p=0.05).

## DISCUSSION

To be effective, NR requires knowledge of the resuscitation algorithm and technical competence in performing PPV and awareness of the need for continual re-evaluation of a neonate's clinical status, including assessment of heart rate. With PRONTO training, as part of the larger AMANAT nurse mentoring interventions in Bihar, an increase was observed in the proportion of PHCs in which PPV was delivered with effective chest rise and heart rate was assessed from mid to post training, consistent with a previous assessment conducted in 2017.[18] However, this initial facility-level improvement was followed by a decline in these same skills over a prolonged training gap. Nevertheless, under the new mentorship model of peer simulation facilitation employed in repeat PRONTO training as part of AMANAT Jyoti, the proportion of PHCs in which heart rate was assessed significantly improved again by mid reassessments with only one NR simulation session.

To maintain technical NR skills, such as PPV, frequent repetition is key. Similar to this study, other studies have shown a decline in NR skills at 1–6 months post training without such repetition.[9–14] Programmes that have achieved skill retention have in common the opportunity for frequent and ongoing NR practice through the time of skill reassessment.[14–16] This study emphasises this need for frequent NR skill practice among frontline providers, who may not have these opportunities in their daily clinical work, despite the 8-month duration of initial PRONTO training within the AMANAT mentoring programme. This need for repetition in NR training was underscored by AMANAT mentors in previous qualitative interviews due to the fact that evidence-based NR practices represent a significant departure from traditional clinical practices in Bihar.[19 20] PPV and heart rate assessment are perhaps the most unfamiliar and technical skills of basic NR, thus it is not surprising that these skills were the ones to degrade over the training gap.[32] In a previous study, AMANAT mentors also identified supply issues, including availability of appropriately sized masks for ventilation bags and watches for keeping time, as additional barriers to PPV and heart rate assessment.[18] Although all necessary supplies were provided during simulation, lack of supply availability during the training gap may have contributed to the observed facility level skill decline by limiting practice opportunities.

Among the other skills assessed, no significant changes were observed at the facility level across all timepoints. It is possible that this was due to inadequate sample size; nevertheless, this pattern is overall consistent with a prior programme assessments and hypotheses regarding this lack of change are discussed in detail elsewhere.[18] It is

important to note that baseline assessments were not conducted prior to initial PRONTO training in AMANAT phases 2–4. This was a deliberate choice to avoid bias from simulation artefact given mentees' lack of familiarity with simulation-based learning. However, based on previous qualitative analyses of mentors' evaluations of mentees' NR skills prior to training,[18] it is likely that the AMANAT midassessments represent a significant improvement compared with true baseline. Therefore, the lack of change observed in many NR skills in this study presumably reflects maintenance of acquired skills rather than lack of skill acquisition. Nevertheless, there is undoubtedly room for continued improvement.

This study had several limitations. As noted above, the small sample size may have precluded identification of significant changes in the performance of all NR skills. Additionally, this analysis assessed skill at a facility level. Due to high staff turnover at PHCs, not all ANM and GNM mentees participating in training under AMANAT Jyoti had previously received PRONTO training under the AMANAT mentoring programme. As previously stated, the exact percentage of AMANAT Jyoti mentees who had received prior training was unknown. Assessing skill retention at the facility level with this limitation risks overestimating skill decline if nurses not previously trained enter PHCs in which training has already occurred. Another limitation was the difference in the beginning of the simulated NR scenario in AMANAT phases 2 and 3, where the scenario started after birth, versus in AMANAT phase 4 and AMANAT Jyoti, where the scenario started with a normal delivery. To reduce the risk of bias, only skills required in both clinical scenarios were assessed, and the time in which those skills were completed was not assessed despite the importance of urgency in NR. Finally, PRONTO simulation training was implemented as part of the AMANAT mentoring programme as previously described. While we are unable to distinguish the individual impact of simulation and mentoring on performance in assessment simulations, this is not dissimilar to other simulation training programmes that involve more components than simulation alone. This assessment is meant to be an assessment of the PRONTO programme within the context of AMANAT mentoring.

The optimal frequency of basic NR simulation training for frontline providers in LMIC settings remains unknown,[33 34] with no current consensus except that it should occur more frequently than annually.[33] It is estimated that 5%–10% of neonates will require basic resuscitation to transition after birth and, of that group, 3%–6% will require bag mask ventilation.[35] For frontline providers, this translates into very few opportunities to practice technical NR skills. Therefore, annual training is likely insufficient. Two recent studies on cardiopulmonary resuscitation (CPR), a similarly rare event requiring competency in technical skills, suggested repeat training at 1-month intervals was optimal.[36 37] In one study, providers were randomised to receive refresher training at 1-month, 3-month, 6-month or 12-month intervals,

and significantly improved performance was observed in the group that received monthly training compared with all other groups. Moreover, there was no difference in skill retention among groups who received refresher training at 3-month, 6-month or 12-month intervals.[37] A monthly training interval is additionally supported by the fact that 1 month is the earliest published evidence of NR skill decline postsimulation training in the LMIC literature.[9] Furthermore, initial PRONTO training within AMANAT mentoring in Bihar occurred at monthly intervals over an 8-month training period and, at this interval, was demonstrated to improve NR skills in simulated and live deliveries.[18]

Monthly training using external trainers or mentors in an LMIC setting presents the unique challenge of sustainability. One possible solution is peer simulation facilitation. In this study, peer simulation facilitation was introduced in AMANAT Jyoti. Programmatically, it was feasible, and early evidence suggests it enabled reacquisition of heart rate assessment skills by the midpoint of training. Although improvement was limited to one NR skill, this occurred with only one peer-facilitated simulation. Peer simulation facilitation has been studied in other low-resource settings including Uganda where peer-facilitated NR practice after an initial HBB training was demonstrated to promote maintenance of NR skills at 6 months post-training when peer facilitators received phone support from HBB trainers.[14] Additionally, a randomised controlled trial in Syria demonstrated noninferiority of peer-facilitated basic life support training, which includes CPR, as compared with professionally facilitated training.[38] Although more research is required into how to optimally train frontline healthcare providers to become experts in both clinical NR skills and simulation facilitation, limited existing evidence suggests peer instruction may offer both a successful and sustainable model of clinical instruction.

## CONCLUSION

PRONTO simulation training implemented in Bihar as part of CARE India's AMANAT mentoring programme in partnership with the local government had a significant, positive impact on the delivery of effective PPV and neonatal heart rate assessment at the facility level. However, after a prolonged training gap, improvements in these skills were lost despite the relatively long duration of initial simulation training. Similar NR skill decline has been observed as early as 1 month post-training in other LMIC settings, concordant with the limited opportunities frontline providers have to practice the technical skills required in NR. More frequent training is necessary, especially in settings where evidence-based NR skills may represent a large shift in clinical practice. In such contexts, a key challenge is the scale and sustainability of any NR training programme, especially if monthly training is indicated. With further research, peer simulation facilitators may be one way to address these issues.

**Author affiliations**
[1]Department of Pediatrics, University of California San Francisco, San Francisco, California, USA
[2]Maternal, Adolescent, Reproductive, and Child Health Centre, London School of Hygiene and Tropical Medicine, London, United Kingdom
[3]Institute for Global Health Sciences, University of California San Francisco, San Francisco, California, USA
[4]PRONTO International, Patna, Bihar, India
[5]College of Nursing, University of Utah, Salt Lake City, Utah, USA
[6]Care India, Patna, Bihar, India
[7]School of Medicine and Department of Obstetrics-Gynecology and Reproductive Sciences, University of California San Francisco, San Francisco, California, USA

**Acknowledgements** The authors would like to thank CARE India as well as the nurse supervisors, nurse mentors and mentees working in the primary health centres throughout the state of Bihar for their hard work implementing and participating in both the Apatkaleen Matritva evam Navjat Tatparta (AMANAT) and AMANAT Jyoti nurse mentoring programmes. Thank you also to the PRONTO Patna-based staff members for facilitating simulation and team trainings and implementing the PRONTO curriculum in both programmes as well as collecting, organising and managing hundreds of simulation videos. Lastly, we would like to thank Dr Hemant Shah and the rest of the CARE India management for their unrelenting efforts to improve conditions for women and newborns in Bihar.

**Contributors** BVH led study design for this manuscript, analysed all simulation videos, performed the data analysis and drafted and revised the manuscript. MMM and HS contributed to study design, literature review, data interpretation and critical manuscript revision. RG, OL and SRC were also involved study design, data interpretation and provided critical manuscript revision. RG additionally provided input in data analysis, and OL additionally provided on the ground oversight of training and local contextualisation. AD, AG and TM were involved in study design, implementation, provided local contextualisation and provided critical input during manuscript revision. DMW was the principal investigator and a major contributor to all aspects of this study and manuscript.

**Funding** This study was supported by the Bill and Melinda Gates Foundation, grant number OPP1112431.

**Disclaimer** The funding body had no role in study design, data collection, analysis, interpretation, manuscript writing or the decision to submit the manuscript for publication.

**Competing interests** DMW and SRC are founding members of PRONTO International and sit on its board of directors. None of the other authors have any competing interests to declare.

**Patient and public involvement** Patients and/or the public were not involved in the design, or conduct, or reporting, or dissemination plans of this research.

**Patient consent for publication** Not required.

**Provenance and peer review** Not commissioned; externally peer reviewed.

**Data availability statement** Data are available on reasonable request. Data are not yet publicly available as the study is ongoing. Requests for data/materials may be made to the corresponding author.

**ORCID iD**
Brennan V Higgins http://orcid.org/0000-0002-8304-6658

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
