## [Reviewer comments · BMJ Paediatrics Open]

ARTICLE DETAILS

TITLE (PROVISIONAL)	A cohort study of neonatal resuscitation skill retention in frontline health care facilities in Bihar, India after PRONTO simulation training
AUTHORS	Higgins, Brennan; Medvedev, Melissa; Spindler, Hilary; Ghosh, Rakesh; Longkumer, Ojungsangla; Cohen, Susanna; Das, Aritra; Gore, Aboli; Mahapatra, Tanmay; Walker, Dilys

VERSION 1 – REVIEW

REVIEWER	Reviewer name: Elizabeth M Molyneux Institution and Country: College of Medicine, Box 360 Blantyre, Malawi or Flat 4, Cap Martin, The Serpentine, Blundellsands, Liverpool L23 6TD Competing interests: I teach on neonatal training courses in Malawi (COIN - care of the young infant and neonate) and in ETAT(+) which Emergency Triage Assessment and Treatment
REVIEW RETURNED	27-Dec-2019

GENERAL COMMENTS	In this paper the authors report the long term results of training in neonatal resuscitation in Bihar India. The training called PRONTO is embedded in a wider programme for maternal and neonatal care (AMANAT) that is running in the same district. Technical skills are known to require frequent retraining and mentorship if they are to be maintained and PRONTO has provided training for 6 days (rather than 2) at the commencement of phases of the project and followed up with regular mentorship. Assessments were made of taught skills at 3 monthly intervals. The study is trying to answer 2 questions: - Does longer training with ongoing mentorship mean skills are maintained? - Who should do the training and mentoring – degree nurse trainers or peer mentors with back up telephonic support. The authors found that even longer initial training and regular supervisory visits did not lead to retention of technical skills; and peer mentorship worked as well as having higher qualified external trainers. Comments: Efforts to reduce perinatal mortality and empower staff working in less than ideal circumstances are to be applauded. Training is important but must be seen to lead to real change and improved outcomes. It is interesting to note that despite their best efforts prolonged training and supervision have not been as effective as hoped. It would help to understand these results if we knew what the staffing levels and cadres are at the primary health centres (PHC) at which deliveries are taking place. How many deliveries are undertaken annually – how many mothers are referred for delivery
---

	elsewhere? Do the midwives have other duties at the PHC that are unrelated to maternal and neonatal care? Do mothers have to pay for medical attention? Are the nursing staff routinely moved to other sections of the PHC or to other centres? Are the nurses paid to attend any trainings? Are they paid to be involved in the PRONTO or AMANET projects? What other similar trainings have the nurses received? (It is not unusual for different international partners to come and train without realising that others have done something similar previously which may confuse health workers rather than reinforce training messages). The text of this paper is hard to read. It would be helpful if much of the methods could be in diagrammatic form. I got bogged down in As and Bs and Cs and A-C etc etc. It is hard to visualise the flow of the study. Even the tables suffer, in my view, from the same problem. The limitations are recognised by the authors – small sample size, unknown number of participants had received previous PRONTO training, not all the set assessments were identical. Nevertheless, the conclusions are interesting and other training teams will be interested in the conclusions of the authors.
--	---

REVIEWER	Reviewer name: Daan Van Brusselen Institution and Country: Ghent University, Belgium Competing interests: None
REVIEW RETURNED	30-Dec-2019

GENERAL COMMENTS	An important conclusion: 'Despite a longer duration of initial training skill deterioration was still observed post-training'. Worth publishing. Conclusion of abstract: "Employing peer simulation facilitators may address the problem of sustainability that arises with increased training frequency." I think this sentence could be used for the conclusion for the article, but not for the conclusion of the abstract, since there is no evidence for it in the article. For the peer trainers during the repeat training almost no changes were detected (see table 2). Same for 'what this study adds': "Nevertheless peer simulation facilitation proved to be an effective way to help frontline providers begin to re-build skills during refresher trainings." There is no clear evidence for this in the article and thus I don't think it should be part of this section. It could be part of the conclusion of the article though. A strength = that the authors excluded the first phase of training from the evaluation because of differences during that initial stage. Something about the 'authorship'. It's a bit of a pity that the first 2 and the last authors are HIC specialists. The Lancet Global Health (The Lancet Global Health. Closing the door on parachutes and parasites. Lancet Glob Health. 2018; 6: e593) have stated last year that they look unfavourably on papers submitted by authors who have done primary research in another country (particularly a low-income or middle- income country) but included no/few authors from that nation.
---

	I can see that there are people from CARE India and Pronto India among the authors as well, but it would have been nice, if they had received a more favourable place. No multivariate statistics were done, but I'm not sure if this is needed for this design. I cannot judge Cochran's Q test, so I'll leave this to the stat reviewer. p3, r 17-18 typo: Bachelor Trained Nurses p3, r25-26, put a comma behind (13,1%) to make the sentence more readable My recommendation: major revision.
--	---

REVIEWER	Reviewer name: Peter Flom Institution and Country: Peter Flom Consulting, USA Competing interests: None
REVIEW RETURNED	10-Jan-2020

GENERAL COMMENTS	I confine my remarks to statistical aspects of this paper. Unfortunately, unless I have misunderstood the design, the analysis chosen was not correct. Rather than look at pairs of time points, the authors should use a multilevel model which lets them look at all the time points together and also makes use of incomplete data. They may want to use a spline of time to model nonlinearities. More specific issues: p 7 I do not understand why time points were lumped together this way. There are 8 time points, not 4. Deleting cases with any missing data is not appropriate. MLMs (as I recommended above) allow use of all the data. Another possibility is multiple imputation. Table 1 should be redone with 8 time points. not 3. Figure 1 is too small to be readable (at least when I printed it) Figures 2 and 3 should be line graphs with time on the x axis, % on the y axis and a line for each skill. And it should have 8 time points.
--

VERSION 1 – AUTHOR RESPONSE

Reviewer: 1

It would help to understand these results if we knew what the staffing levels and cadres are at the primary health centres (PHC) at which deliveries are taking place.

Thank you for this suggestion. We have answered your specific questions below and added these details to the methods section of the manuscript under "Study setting and population" (page 4).

How many deliveries are undertaken annually?

Approximately 1.3 million deliveries are conducted annually in the PHCs mentored under AMANAT/AMANAT Jyoti (n=353). The exact delivery load per PHC varies by PHC location.

How many mothers are referred for delivery elsewhere?

Approximately 10% of mothers are referred from PHCs to higher-level facilities for delivery.

Do the midwives have other duties at the PHC that are unrelated to maternal and neonatal care?

Yes, midwives have a variety of other duties in addition to labour and delivery duties. These include other clinical obligations, including the emergency care and outpatient care sections of the PHC, as well as clerical tasks such as taking supply inventory and completing maternity registers.

Do mothers have to pay for medical attention?

In 2019, the average out-of-pocket expenditure was 1788 Indian rupees which is equivalent to roughly 25 United States dollars. The majority of these expenditures were for diagnostic tests.

Are the nursing staff routinely moved to other sections of the PHC or to other centres?

Nursing staff are not routinely moved around. However, they do travel to smaller community sub-centres twice per week to perform routine immunisations.

Are the nurses paid to attend any trainings? Are they paid to be involved in the PRONTO or AMANAT projects?

ANM/GNMs are not paid to participate in PRONTO or AMANAT trainings or other trainings.

What other similar trainings have the nurses received?

Some ANM/GNM mentees had participated in other governmental classroom-based trainings on topics including basic newborn care and neonatal resuscitation. They had not received any prior simulation training.

The text of this paper is hard to read. It would be helpful if much of the methods could be in diagrammatic form. I got bogged down in As and Bs and Cs and A-C etc etc. It is hard to visualise the flow of the study. Even the tables suffer, in my view, from the same problem.

We acknowledge this intervention is complex as it spans 5 years and includes two different training models under AMANAT and then AMANAT Jyoti. We have tried to make the methods section more clear. Additionally, we have re-done Figure 1 to encompass additional aspects of the methods section. Finally, we removed the nomenclature, "timepoints A, B, C, D," and instead have labelled the timepoints to improve clarity, as follows: "AMANAT mid assessment," "AMANAT post assessment," "AMANAT Jyoti baseline re-assessment," and "AMANAT Jyoti mid re-assessment." We have made these changes throughout the manuscript as well as in the tables.

Reviewer: 2

Conclusion of abstract: "Employing peer simulation facilitators may address the problem of sustainability that arises with increased training frequency." I think this sentence could be used for the conclusion for the article, but not for the conclusion of the abstract, since there is no evidence for it in the article. For the peer trainers during the repeat training almost no changes were detected (see table 2).

Thank you. To avoid over stating our conclusions in the abstract we have changed the conclusion of this section to, "Employing peer simulation facilitators may enable such increased training frequency."

Same for 'what this study adds': "Nevertheless peer simulation facilitation proved to be an effective way to help frontline providers begin to re-build skills during refresher trainings." There is no clear evidence for this in the article and thus I don't think it should be part of this section. It could be part of the conclusion of the article though.

We agree this comment warrants a more nuanced discussion in the article but is not appropriate for the initial summary bullet points. We have changed this statement to, "Refresher trainings were possible with peer simulation facilitators and with further study could be an effective model for providing more frequent simulation training in frontline facilities."

Something about the 'authorship'. It's a bit of a pity that the first 2 and the last authors are HIC specialists. The Lancet Global Health (The Lancet Global Health. Closing the door on parachutes and parasites. Lancet Glob Health. 2018; 6: e593) have stated last year that they look unfavourably on papers submitted by authors who have done primary research in another country (particularly a low-income or middle- income country) but included no/few authors from that nation. I can see that there are people from CARE India and Pronto India among the authors as well, but it would have been nice, if they had received a more favourable place.

Collaboration with CARE India for the AMANAT and AMANAT Jyoti intervention made the PRONTO simulation training in Bihar possible. This manuscript is an assessment of PRONTO specific activities within the context of AMANAT. However, since this manuscript is based solely on PRONTO simulation data and primary analysis and drafting of the manuscript was done by the PRONTO team, those contributions are reflected in the chosen author order. We followed the ICMJE criteria to decide authorship and the order was decided in consultation with Dr. Tanmay Mahapatra (second to last author), who leads the CARE Continuous Monitoring and Learning team in Bihar, India (not a HIC specialist). We nevertheless rely on input from our CARE India colleagues in data interpretation and representation and feel strongly that they are included as authors on all PRONTO papers. Drs. Aritra Das and Aboli Gore are members of the CARE team and Ms. Ojungsangla Longkumer is part of the PRONTO India team. Conversely, for assessments of the AMANAT program, CARE India has taken the lead.

No multivariate statistics were done, but I'm not sure if this is needed for this design. I cannot judge Cochran's Q test, so I'll leave this to the stat reviewer.

Please see responses to reviewer 3 below.

p3, r 17-18 typo: Bachelor Trained Nurses

Corrected. Thank you.

p3, r25-26, put a comma behind (13,1%) to make the sentence more readable

We have moved all numbers to the end of that sentence to make it more readable.

Reviewer: 3

Unfortunately, unless I have misunderstood the design, the analysis chosen was not correct. Rather than look at pairs of time points, the authors should use a multilevel model which lets them look at all the time points together and also makes use of incomplete data.

In brief, this study looks at NR skills at a facility level at 4 timepoints—AMANAT mid, AMANAT post, AMANAT Jyoti baseline, and AMANAT Jyoti mid. (See below for a discussion of why these timepoints were chosen.) With these 4 timepoints we attempt to fulfill two objectives, 1) assess retention of NR skills after PRONTO simulation training over a training gap and 2) assess skill re-acquisition with peer simulation training. We have better laid out these two separate objectives at the end of the introduction. AMANAT mid and AMANAT post are two time points of the same intervention program (“AMANAT”), while AMANAT Jyoti baseline and AMANAT Jyoti mid are two time points of a separate intervention program (“AMANAT Jyoti”). Although the broad objective of the two training programs is the same (i.e., to improve quality of obstetric and neonatal care services in government health facilities in Bihar), there is a salient difference in the way the two interventions were delivered. Further, there is also a difference in training dose- while AMANAT training was once per month, AMANAT Jyoti training was approximately once every quarter. Therefore, it would be qualitatively inappropriate to group all of the timepoints from the two programs together using multilevel models. We agree with the reviewer that this will add power to the study. However, the bigger question is, does this level of increased complexity in the analysis enrich the results and add substantial value to the manuscript.

We believe that the statistical methods used were correct to fulfill the two stated objectives. Objective 1 is met in table 1. In table 1, we used the Cochran’s Q test to compare NR skills matched by facility at three time points (AMANAT mid, AMANAT post, AMANAT Jyoti baseline). We then used the McNemar’s test to better understand change between all individual timepoints for any skill in which the Cochran’s Q test was significant. This avoided assumptions about any trends over the 3 timepoints. We were able to assess skill acquisition from AMANAT mid to AMANAT post and then assess for skill retention from AMANAT post to AMANAT Jyoti baseline.

Table 2 fulfills objective two. In table 2, McNemar’s test was used to assess the difference between matched frequencies at two timepoints—AMANAT Jyoti baseline and AMANAT Jyoti mid.

We agree that multilevel models will make use of incomplete data. Therefore, we have performed multilevel modeling and added it to the paper as a supplemental table. Please see the “analysis” section of the results for full details about statistical methods used. Briefly, we used a mixed effect logistic regression model with random intercept for facility because time points are clustered within facilities. Results using multilevel models were consistent with our previous analysis when models converged. In case of “Heart rate assessed” the models did not converge. So, if we make these our main results, we will lose key information.

We have chosen to keep the original analysis and add the multilevel modeling as a supplement because we believe that adding this level of complexity only makes the results more difficult to interpret without adding additional value to the manuscript. Additionally, looking at skill gain/loss by unit of time in the multilevel model (1 month was chosen for ease of interpretation) does not make sense based on the intervention. During AMANAT and AMANAT Jyoti the PRONTO training

frequency was different so choosing a uniform unit of time that is applicable to both interventions is not possible. It makes more sense to simply assess skills at the chosen timepoints (AMANAT mid, AMANAT post, AMANAT Jyoti baseline, AMANAT Jyoti mid) rather than make assumptions about skill gain per month, which may not be consistent with patterns of learning and is not as programmatically relevant. Additionally, with the multilevel models we lose important information on heart rate assessment due to model non convergence.

As elaborated earlier, looking at all timepoints all together using multilevel modeling is also not programmatically relevant. We expect different changes in skill depending on when skills are assessed -- skill improvement from mid to post during the AMANAT phase of training, possible skill attrition during the training gap between AMANAT and AMANAT Jyoti, skill re-acquisition from AMANAT Jyoti baseline to mid. Furthermore, as we are not examining association, we did not adjust for confounding. To adjust for potential confounding, regression models would be required.

They may want to use a spline of time to model nonlinearities.

For most outcomes there is no evidence of non-linearity warranting spline, particularly when using two timepoints, appropriate because of the nature of the two (AMANAT and AMANAT Jyoti) intervention programs. When we consider all time points, there is evidence of non-linearity in PPV with chest rise, chest rise in less than or equal to 30 seconds, and heart rate assessed- please see figures below.

Nevertheless, we have accounted for any nonlinear change in skills by separating timepoints and performing different multilevel models for each period of interest as follows: AMANAT mid to AMANAT post, AMANAT post to AMANAT Jyoti baseline, AMANAT mid to AMANAT Jyoti baseline, AMANAT Jyoti baseline to AMANAT Jyoti mid. Separating the analysis into these intervals is programmatically more relevant as we expect different patterns of skill acquisition or loss based on what time interval is being assessed as explained above.

p 7 I do not understand why time points were lumped together this way. There are 8 time points, not 4.

This analysis is a facility based analysis. Facilities only participated in one AMANAT training phase and then received PRONTO training for a second time during AMANAT Jyoti (we have attempted to clarify this in the methods section as well as in the improved Figure 1). Thus for each facility skills were only assessed at 4 timepoints- AMANAT mid, AMANAT post, AMANAT Jyoti baseline, AMANAT Jyoti mid.

Deleting cases with any missing data is not appropriate. MLMs (as I recommended above) allow use of all the data. Another possibility is multiple imputation.

As stated above multilevel models have been conducted to account for missing data and included as a supplemental figure. Results were consistent.

Table 1 should be redone with 8 time points. not 3.

Please see comment above regarding timepoints. Table 1 is assessing skill acquisition with initial training and retention over a training gap thus the timepoints for assessment were 1) AMANAT mid, 2) AMANAT post, 3) AMANAT Jyoti baseline for each PHC.

Figure 1 is too small to be readable (at least when I printed it). Figures 2 and 3 should be line graphs with time on the x axis, % on the y axis and a line for each skill. And it should have 8 time points.

We have decided to remove the figures, as they do not add additional information to the manuscript but rather only visually depict the results of tables.

VERSION 2 – REVIEW

REVIEWER	Reviewer name: Peter Flom Institution and Country: Peter Flom Consulting, USA Competing interests: None
REVIEW RETURNED	17-Feb-2020

GENERAL COMMENTS	I mostly confine my remarks to statistical aspects of this paper. My main issue is that I think the multilevel model described on p. 8 (and its results) should be the main focus of the paper, and the analysis described on p. 7 should be minimized or even removed. More specific comments: p 3, line 26 There should be two p values, shouldn't there? line 30 Insert "significant" between "no" and "change"
---

	p 5 to 7 I was somewhat confused. The design is complex but it needs to be more clearly described. Who was assessed when? Was it the same people or the same sites or what? It would be much more powerful to be able to assess the same nurses over time. p 7 Lines 44-54 This is the analysis I think should be removed. Removing missing data is going to bias the results and MLMs allow more interesting comparisons. Supplementary table 1 should be in the main text.
--	---

REVIEWER	Reviewer name: Daan Van Brusselen Institution and Country: University of Ghent, Belgium Competing interests: None
REVIEW RETURNED	17-Feb-2020

GENERAL COMMENTS	I think the article has improved significantly (esp. multivariate stats as well I think, but I leave that to the stats reviewer). Nevertheless, even if I want to believe that employing peer simulation facilitators may be one way to address the issue of poor retention of NR skills, I still believe this is overstated in the abstract and conclusion. Unless I'm mistaken, the only evidence there is for this in the article, is 'heart rate assessment' in table 2 (only p-value <0.05). Maybe to be changed to: "So far there is not a lot of evidence that employing peer simulation facilitators may enable such increased training frequency." Up to the editor... For the abstract, please review this sentence in the 'What this study adds': "Refresher trainings were possible with peer simulation facilitators and with further study could be an effective model for providing more frequent simulation training in frontline facilities". I somehow understand what you mean, but this sentence is not clear.
--

VERSION 2 – AUTHOR RESPONSE

Reviewer: 1

I mostly confine my remarks to statistical aspects of this paper. My main issue is that I think the multilevel model described on p. 8 (and its results) should be the main focus of the paper, and the analysis described on p. 7 should be minimized or even removed.

We have added the multilevel model to the main paper and described it more fully in the results section following our descriptive analysis as originally presented. We hope this balanced approach is acceptable to the reviewer and the editor. We do not think that the original analysis should be completely removed for several reasons. First, with the multilevel models we lose important information about heart rate assessment due to model non convergence. Second, the findings of the multilevel model and our original analysis are concordant and thus the impact of missing data in the Cochran's Q/McNemar's analysis does not appear to be significant enough to warrant complete removal. Third, we believe the original analysis is more comprehensible to readers with little statistical background and that alone the multilevel model would result in a paper that is difficult to interpret. Fourth, we believe our original analysis is more programmatically relevant. This programmatic relevance is due first to the fact that simulation dosages were different across AMANAT and AMANAT

Jyoti as described in the methods (AMANAT training was once per month, AMANAT Jyoti training was approximately once every quarter; they were also conducted by different cadres of trainers). Thus, there is a difficult choice to make between building uniform statistical models across time points and assessing the intervention the way it was actually implemented with optimal programmatic relevance. We believe evaluating skills at the chosen assessment timepoints (AMANAT mid, AMANAT post, AMANAT Jyoti baseline, AMANAT Jyoti mid) as done in the descriptive analysis rather than forcing a uniform unit of time on the analysis better captures both patterns of human learning and the way simulation training was actually implemented across the study periods.

More specific comments:

p 3, line 26 There should be two p values, shouldn't there?

Both p-values here are the same, 0.03. We have added the word both to clarify.

line 30 Insert "significant" between "no" and "change"

The word "significant" has been added. Thank you.

p 5 to 7 I was somewhat confused. The design is complex but it needs to be more clearly described. Who was assessed when? Was it the same people or the same sites or what? It would be much more powerful to be able to assess the same nurses over time.

Thank you for this comment. We have re-structured the former "AMANAT and PRONTO interventions" section in the methods into two sub-sections-- 1) "AMANAT and PRONTO interventions" and 2) "Simulations and assessments" to try to improve clarity.

The specific answers to your questions have been included and clarified in the "Simulations and assessments" section. This is a facility-based analysis. One assessment simulation was conducted per facility at each assessment time point and the individual participants were chosen at random. It was not possible to assess the same nurses over time given their varied obligations within and outside the PHC (described in the second paragraph of "Study setting") as well as staff turnover during the 5-year time period.

p 7 Lines 44-54 This is the analysis I think should be removed. Removing missing data is going to bias the results and MLMs allow more interesting comparisons.

Please see response to first comment above.

Supplementary table 1 should be in the main text.

The supplementary table has been added to the main text as table 3.

Reviewer: 2

I think the article has improved significantly (esp. multivariate stats as well I think, but I leave that to the stats reviewer). Nevertheless, even if I want to believe that employing peer simulation facilitators may be one way to address the issue of poor retention of NR skills, I still believe this is overstated in the abstract and conclusion. Unless I'm mistaken, the only evidence there is for this in the article, is 'heart rate assessment' in table 2 (only p-value <0.05).

Maybe to be changed to: "So far there is not a lot of evidence that employing peer simulation facilitators may enable such increased training frequency." Up to the editor.

Thank you for highlighting this point. We agree our findings regarding peer simulation facilitators should be interpreted conservatively not only because of the limited statistical significance but also because only one peer led simulation occurred between AMANAT Jyoti baseline re-assessment and AMANAT Jyoti mid re-assessment. Thus, to avoid overstatement in the abstract, we have changed the final sentence to, "Very limited evidence suggests peer simulation facilitators may enable such increased training frequency but further study is required."

In the discussion, we have added a sentence to the final paragraph to clarify the context and limitations of our data. Because our data is limited, we cite two other studies (citation 14 and 38) focused on peer simulation facilitation. These two studies support the success of peer facilitated simulation. Thus, our conclusion in the discussion is that further research is necessary but that peer simulation facilitators may be a successful model. We have changed the final sentence of the conclusion to echo this and to avoid overstatement.

For the abstract, please review this sentence in the 'What this study adds': "Refresher trainings were possible with peer simulation facilitators and with further study could be an effective model for providing more frequent simulation training in frontline facilities". I somehow understand what you mean, but this sentence is not clear.

We have edited this sentence to clarify. It now reads, "Refresher trainings were possible with peer simulation facilitators, but demonstrated limited skill improvement among nurse mentees."